# Analysis of reflections in GNSS radio occultation measurements using the phase matching amplitude

Thomas Sievert[1], Joel Rasch[2], Anders Carlström[3], and Mats I. Pettersson[1]

[1]Blekinge Institute of Technology, Karlskrona, Sweden
[2]Molflow, Gothenburg, Sweden
[3]RUAG Space AB, Gothenburg, Sweden

*Correspondence to:* Thomas Sievert (thomas.sievert@bth.se)

**Abstract.** It is well-known that in the presence of super-refractive (SR) layers in the lower troposphere inversion of GNSS radio occultation (RO) measurements using the Abel transform yields biased refractivity profiles. As such it is problematic to reconstruct the true refractivity from the RO signal. Additional information about this lower region of the atmosphere might be embedded in reflected parts of the signal. To retrieve the bending angle, the phase matching operator can be used. This operator produces a complex function of the impact parameter, and from its phase we can calculate the bending angle. Instead of looking at the phase, in this paper we focus on the function's amplitude. The results in this paper show that the signatures of surface reflections in GNSS RO measurements can be significantly enhanced when using the phase matching method by processing only an appropriately selected segment of the received signal. This signature enhancement is demonstrated by simulations and confirmed with ten hand-picked MetOp-A occultations with reflected components. To validate that these events show signs of reflections, radio holographic images are generated. Our results suggest that the phase matching amplitude carries information that can improve the interpretation of radio occultation measurements in the lower troposphere.

## 1 Introduction

GNSS radio occultation (RO) is a technique used for sounding the Earth's atmosphere. Assuming spherical symmetry of the atmosphere, the bending angles of GNSS signals passing through the atmosphere can be found and assimilated into numerical weather prediction (NWP) systems. The bending angle measurements contain valuable information due to their relation to the atmosphere's refractivity, which can yield information about humidity, temperature and pressure (e.g., Kursinski et al., 2000; Yunck et al., 2000). The geometry of transmitter, Earth, and receiver, as well as the short wavelength of the signals, result in a measurement with high vertical resolution. In many RO events, the instrument in orbit receives reflected components of the signal as well as direct ones. Boniface et al. (2011) have shown that reflected signals contain meteorological information. A method to detect these reflected components was suggested by Hocke et al. (1999) and has later been used on real data and shown to work (e.g., Beyerle et al., 2002; Pavelyev et al., 2002). This method uses a radio hologram generated by subtracting a ray traced reference field from the received signal. An effort to flag occultation events where reflections are present is described by Cardellach and Oliveras (2016), based on a supervised learning approach classifying such radio holographic images. It has since then been employed by the Radio Occultation Meteorology Satellite Application Facility (ROM SAF) to flag millions

of occultation events. Cardellach and Oliveras also investigate whether knowledge of these reflections can improve the quality of RO data, however Healy (2015) concludes that a binary reflection flag is probably inappropriate for assimilation purposes. Gorbunov (2016) proposes a technique based on the canonical transform to retrieve bending angle profiles of reflected rays, and achieves a good agreement with the ROM SAF database.

When processing a RO signal, we use the phase matching (PM) operator (Jensen et al., 2004), which outputs a complex function of impact parameter whose phase is proportional to the signal's bending angle. Although the amplitude of this function may contain valuable information as well, it has not been appropriately investigated.

In this paper, we compare the phase matching amplitudes of simulated measurements to real measurements made by the GRAS instrument aboard MetOp-A. We demonstrate signatures corresponding to surface reflections, and truncate the received signal to distinguish them from the much stronger signatures of the direct components. We motivate this truncation using simulated data and a simple geometric model for where reflected components are expected to appear in the signal, and we provide radio holographic images as a means of validation. Finally we discuss the difference in structure between simulated and real signals, as well as the potential future uses of the PM amplitude. In the appendix we motivate that the PM operator admits reflected signal components.

## 2  Phase matching

Jensen et al. define PM as an operator that transforms a complex signal of time $u(t)$ to impact parameter space:

$$U(a) = \int_{t_{min}}^{t_{max}} u(t) \exp\Big(-ik_0 S(t,a)\Big) dt \tag{1}$$

Here, $k_0$ is the wave number in vacuum associated to the carrier frequency of the GNSS transmitter, $S(t,a)$ is the optical path length of a model ray path, and $a$ is impact parameter. The derivative of $arg(U(a))$ with respect to $a$ is proportional to the bending angle. As this integral is defined for any impact parameter $a$, it is important to determine at which $a$ the function does not contain relevant information anymore. The lowest $a$ (which we call $a_{min}$ in this paper) corresponds to a ray that is tangential to the Earth's surface. To make sure that all information in the signal is mapped to impact parameter space, we compute the $U$ for impact parameter values going all the way to the Earth's surface. This also ensures that we include reflected rays in the $U$ function. It is not obvious that the PM method should work for reflected rays, as the geometrical model ray path is constructed for a direct ray, but in the appendix it is shown that the standard phase matching technique admits reflected rays as well, provided the Earth surface is smooth.

## 3  A model for reflected rays

In Fig. (1) we use data from two real MetOp-A measurements to demonstrate that some of the features we see in the amplitude for the complex function $U$ are caused by reflections. By overlaying the predicted straight line tangent altitude (SLTA) for the direct rays (blue line), and reflected rays (red line) as a function of impact height the reflection is illustrated. To generate these

SLTA plots we use the co-located ECMWF refractivity profiles. The black lines show the amplitude of the $U$ function when passing segments of the signal to the PM operator using a 10 km sliding window. The relationship between reflected bending angle and impact parameter is well-known (see e.g., Pavelyev et al., 2011; Gorbunov, 2016) and can be described as

$$\alpha(a) = -2a \int\limits_{R_E n(R_E)}^{\infty} \frac{1}{\sqrt{r^2 n^2 - a^2}} \frac{d \ln n}{dr} dr - \pi + 2 \arcsin \left( \frac{a}{R_E n(R_E)} \right), \tag{2}$$

where $\alpha$ is the bending angle, $R_E$ the Earth radius of curvature, and $n$ the refractive index as a function of radius. The bending angle of a direct ray is given by the same expression without the last two terms. The integral can be evaluated numerically using a number of techniques, and we employ the method described in (Rasch, 2014). The SLTA for fixed values of the LEO and GNSS orbital radii is given by

$$SLTA = \frac{r_L r_G \sin \theta}{\sqrt{r_L^2 + r_G^2 - 2 r_L r_G \cos \theta}} - R_E, \tag{3}$$

where $r_L$ and $r_G$ are the LEO and GNSS orbit radii, and $\theta$ is the separation angle between the satellites, given by

$$\theta = \pi + \alpha - \arcsin \left( \frac{a}{r_L} \right) - \arcsin \left( \frac{a}{r_G} \right). \tag{4}$$

## 4   MetOp-A data

The data from occultation events is collected from the COSMIC Data Analysis and Archive Center (CDAAC) web interface, specifically day 2007.274 with the metopa2016 designation, indicating reprocessed measurements from MetOp-A. The signal amplitude, excess phase and orbit data needed for PM are all found in the atmPhs files. In these files, the orbit coordinates are given with the Earth's center of mass as the point of origin. To fulfil the assumption of local spherical symmetry, we translate the coordinates so that they instead consider the center of curvature of the Earth at the occultation point. This is done by collecting center of curvature data from the corresponding atmPrf files. As the atmPrf files contain bending angle and impact height values, these were used to control the accuracy of the PM implementation.

For simulating a GNSS signal as it passes through the atmosphere, a wave optics propagator (WOP) is used with the multiple phase screen technique (see e.g., Benzon et al., 2003; Benzon and Gorbunov, 2012; Rasch, 2014). The WOP uses a simpler, two-dimensional geometry where the GNSS transmitter is stationary. The three-dimensional geometry data in the atmPhs files is thus projected onto a plane where the LEO orbit is defined by the separation angle. Both transmitter and receiver are assigned a constant radius, and the Earth is assigned the radius of curvature for the particular event. The separation angles can be modified to simulate an occultation event tracked to a lower SLTA than its corresponding measurement. For an atmosphere, the high-resolution, co-located refractivity profile from ECMWF also provided by CDAAC is used (echPrf files). These profiles do not go all the way to the ground - the last bit is extrapolated linearly. To simulate surface reflections, we set the electromagnetic

field to 0 on all parts of the phase screens that lie inside the Earth. Although it is not clear whether this method has a solid physical basis it appears to give quite accurate results, and is routinely used in WOP simulations (Gorbunov, 2016; Levy, 2000).

As a frame of reference, radio holographic images are produced by constructing a "beating function" from the excess phase,
signal amplitude and a reference field, and using the short-time fast Fourier transform with a sliding window, similar to Boniface et al. (2011).

## 5   Surface reflections

In the lowest part of an occultation (SLTA around $-80$ km, impact height around 2.1 km), where the signal becomes shadowed by the Earth, the magnitude of $U$ also decreases. The lowest direct ray becomes diffracted by the Earth's surface and gradually
decreases in magnitude over a region corresponding to the Fresnel zone size, which is seen clearly in simulated data, and frequently less clearly in measured data. This transition occurs over a few hundred meters (Kursinski et al., 2000), around $SLTA_{min}$, and $a_{min}$, whose values are determined by the Earth radius, and the refractive index at ground. Quite often when the signal is lost at an SLTA value that is substantially higher than $SLTA_{min}$ we see a spike in $|U|$ around $a_{min}$. This spike is most easily explained as a reflection. If tracking of the signal goes all the way to the surface this spike overlaps with the direct
signal and is obfuscated. In Fig. (1) we see that the reflected signal and the direct signal coincide around $a_{min}$ (approximately at 2.1 km). For the left panel this occurs at the deepest SLTA (around $-90$ km), and for the right panel this again occurs around $-90$ km, but the deepest part of the signal goes to $-150$ km, and occurs at an impact height around 2.8 km, which corresponds to a region of strong refractivity gradients. The value for SLTA where the direct and reflected signals join we call $SLTA_{min}$. By cutting the signal well above $SLTA_{min}$ at an SLTA around $-50$ km it is clear from Fig. (1) that we would only keep
the upper parts of the reflected and direct signals, and completely remove the signal parts containing the diffraction signature around $a_{min}$, and the deep signals caused by strong refractivity gradients. In Fig. (2) we can see the same thing. The abrupt decrease in $|U|$ around an impact height of 2.1 km corresponds to the point where the reflected and direct signals join, and the direct signal becomes diffracted. By cutting the signal at $-65$ km the lower parts of the reflected and direct signals are removed, and the reflection signature becomes clearly separated from the direct signal. The reflection spike appears a small
distance below $a_{min}$. This small distance depends on where the signal is truncated, which is illustrated by Fig. (1).

The nature of the PM operator is such that sharp discontinuities in the signal introduce significant noise in the $|U|$ function. To avoid these, we taper the signals using a one-sided Tukey window (Bloomfield, 2004). In the case of the sliding window used in Fig. (1), we use a two-sided window. The noise generated by omitting any sort of tapering is demonstrated in Fig. (2), where the simulated signal ends abruptly.

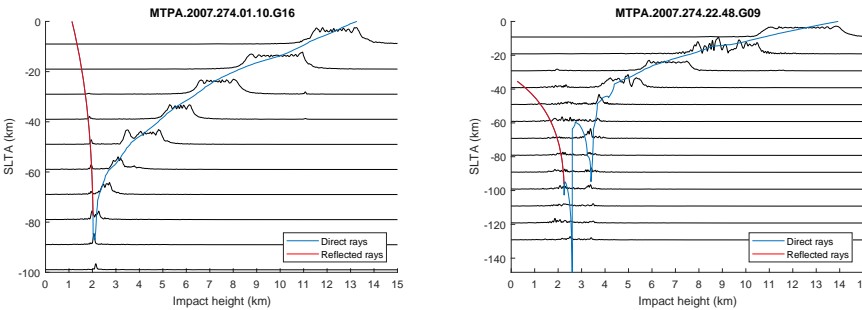

**Figure 1.** Using a sliding window for the signal yields PM amplitudes that correspond to the reflection model. Left shows a case with weak refractive gradients, right shows a case with strong refractive gradients.

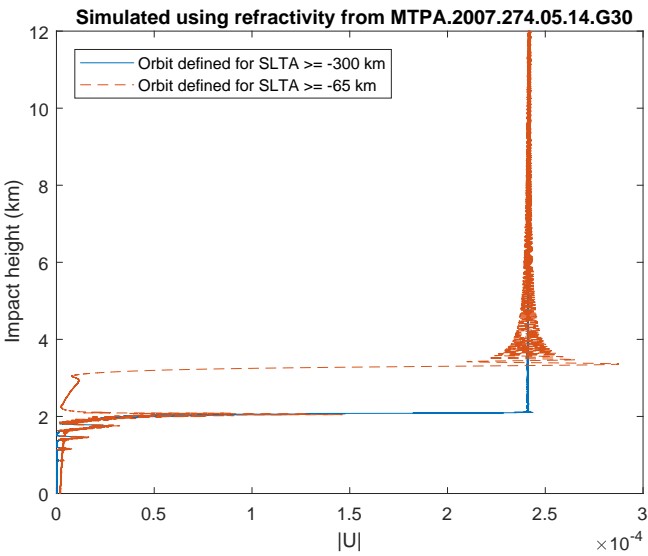

**Figure 2.** $|U|$ for a simulated signal that is propagated to a shallow orbit, and one that is propagated to a deeper orbit (blue). For illustration the truncated signal is not tapered, producing dramatic oscillations around $4$ km.

## 6 Results

We present ten cases where PM is performed on real signals alongside their simulated counterparts based on ECMWF's co-located refractivity profiles. These cases are classified in categories 1 through 4 based on the sharpest gradient above 100 m, in the same manner as the reference dataset from ECMWF (Healy, 2012), where category 4 have the sharpest gradients. Overall, the structure of $|U|$ is similar to that of a step function, both in simulations and using real data. By truncating the signals at an appropriate SLTA we can distinguish previously hidden reflection spikes in $|U|$. For this study we pick the altitude of truncation qualitatively. Figures (3) through (12) show the received signal (first from the left), $|U|$ for a simulated signal (second from the

left), $|U|$ for the received signal (third from the left), and a radio hologram generated from the signal (fourth from the left). The first, second and third plot from the left are color coded so that the blue plots describe the complete signals, and the red plots describe the truncated signals. All the truncated signals have been tapered using the aforementioned Tukey window to avoid introducing additional noise.

Simulations on cases classified as category 3 and 4 - shown in Fig. (8) through (12) - give rise to a sharp, negative spike at an impact height corresponding to the sharp gradient in the refractivity profile. This structure cannot be found in the real data. Moreover, the real data shows a high level of noise that is not found in simulated data.

We note that there are peculiar oscillating structures in the real data. Particularly in Fig. (4) at 6 km, Fig. (7) at 4 and 6 km, Fig. (8) at 7.5 km, Fig. (11) at 5 and 9 km, and in Fig. (12) at 5 and 9 km. These oscillating structures are not found in the simulated data.

Furthermore we note that in all cases but Fig. (11), the reflection spike appears some distance below the $a_{min}$ calculated from the ECMWF profiles.

While these ten hand-picked cases all clearly contain reflected components, there are several measurements in which the reflected parts cannot be distinguished. This is typically either because the measurement was not deep enough, or because $|U|$ was too noisy. When the measurements are sufficiently deep, and the noise level of $|U|$ is low, there are very clear reflection spikes. This is corroborated by the radio holographic images, who also show very clear reflection patterns in those cases. The bright yellow trail around 0 Hz is the direct signal, whereas the more faint trail going off to negative frequencies (and appearing again at positive frequencies due to aliasing) is the reflected signal. The signal has everywhere been divided by the average signal amplitude in the window. This is done to make the reflection and the direct signal for low SLTA stand out against the strong signature of the direct signal for high SLTA. This also often causes the hologram to give the appearance of containing strong broadband noise for low SLTA, but it is actually the amplification of weak noise.

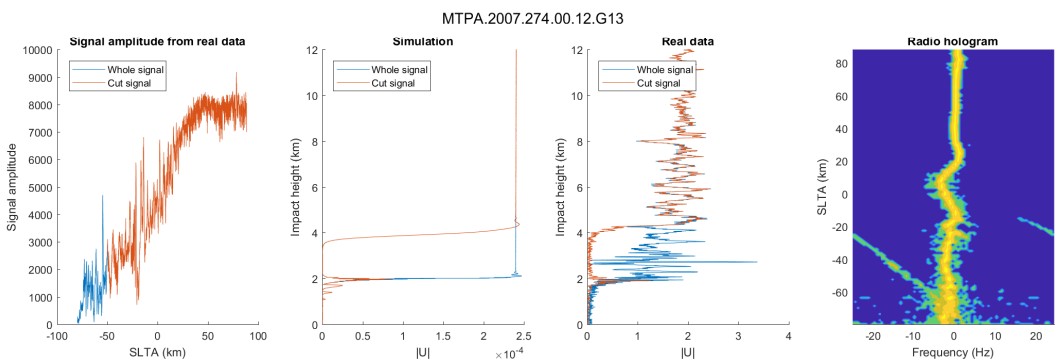

**Figure 3.** An event classified into category 1. $SLTA_{min}$ is at approximately $-73$ km, and the signal is truncated at $SLTA = -50$ km. The reflection spike is located at an impact height of 1919 m, and $a_{min}$ is located at an impact height of 2017 m.

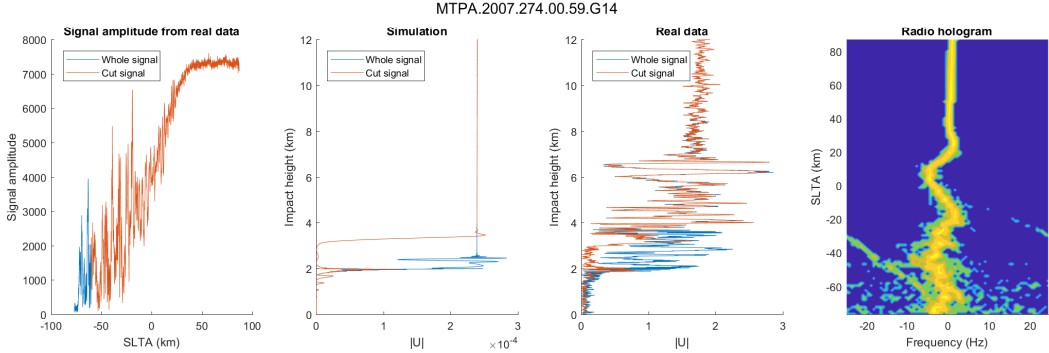

**Figure 4.** An event classified into category 1. $SLTA_{min}$ is at approximately $-73$ km, and the signal is truncated at $SLTA = -60$ km. The reflection spike is 1881 m, and $a_{min}$ is 1992 m.

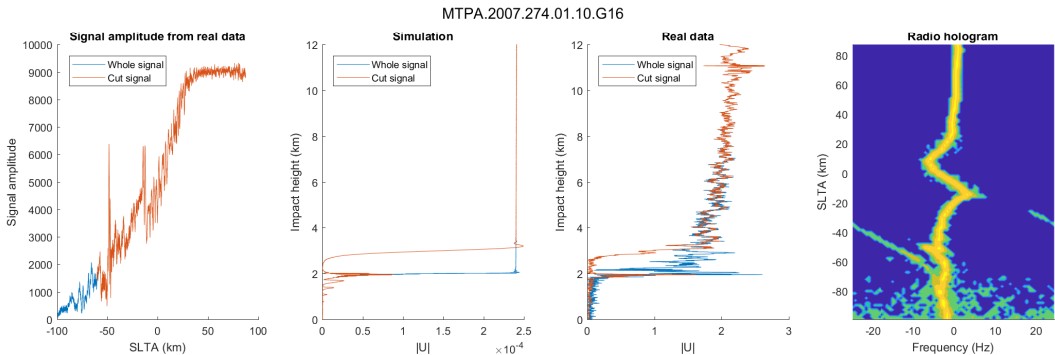

**Figure 5.** An event classified into category 1. $SLTA_{min}$ is at approximately $-86$ km, and the signal is truncated at $SLTA = -60$ km. The reflection spike is located at an impact height of 1948 m, and $a_{min}$ is 2008 m.

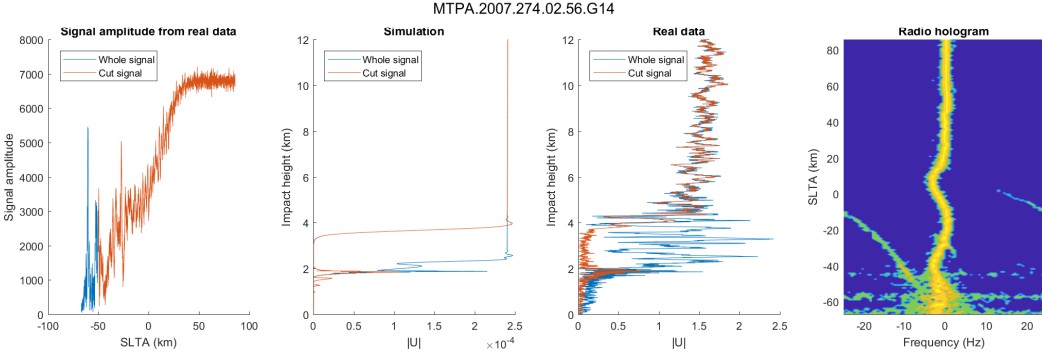

**Figure 6.** An event classified into category 1. $SLTA_{min}$ is at approximately $-65$ km, and the signal is truncated at $SLTA = -50$ km. The reflection spike is located at an impact height of 1849 m, and $a_{min}$ is 1888 m.

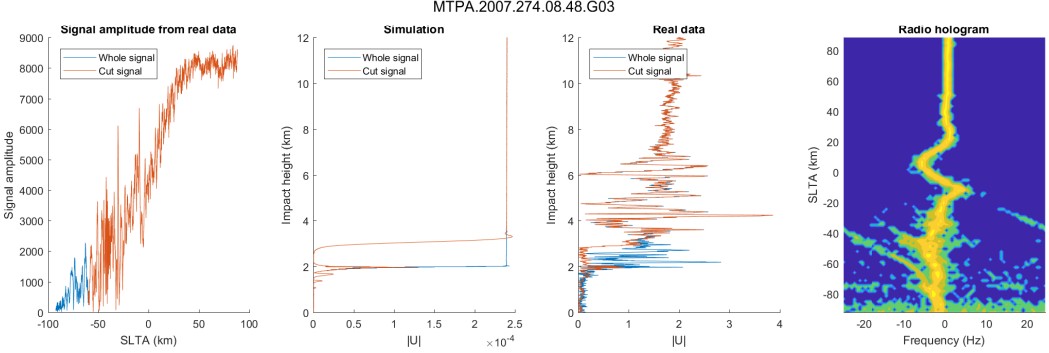

**Figure 7.** An event classified into category 1. $SLTA_{min}$ is at approximately $-72$ km, and the signal is truncated at $SLTA = -60$ km. The reflection spike is located at an impact height of 1940 m, and $a_{min}$ is 1993 m.

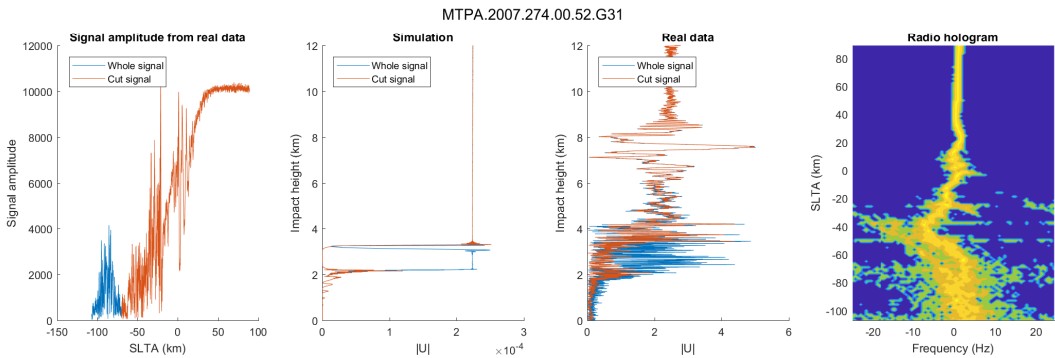

**Figure 8.** An event classified into category 3. $SLTA_{min}$ is at approximately $-87$ km, and the signal is truncated at $SLTA = -70$ km. The reflection spike is located at an impact height of 2075 m, and $a_{min}$ is 2204 m.

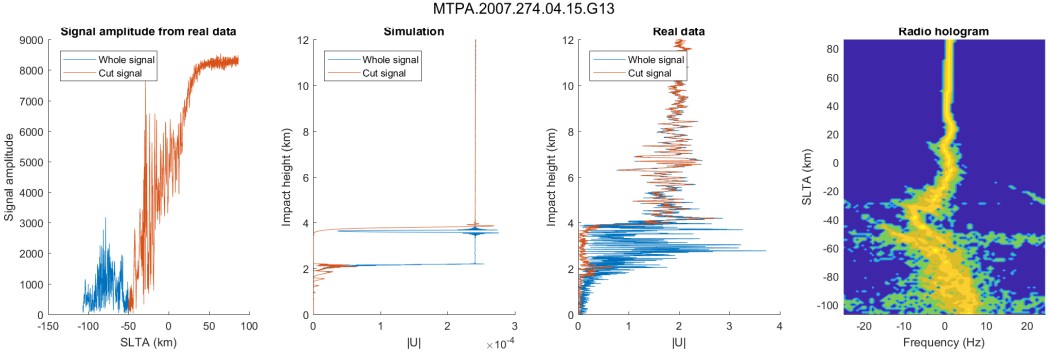

**Figure 9.** An event classified into category 3. $SLTA_{min}$ is at approximately $-84$ km, and the signal is truncated at $SLTA = -50$ km. The reflection spike is located at an impact height of 2057 m, and $a_{min}$ is 2182 m.

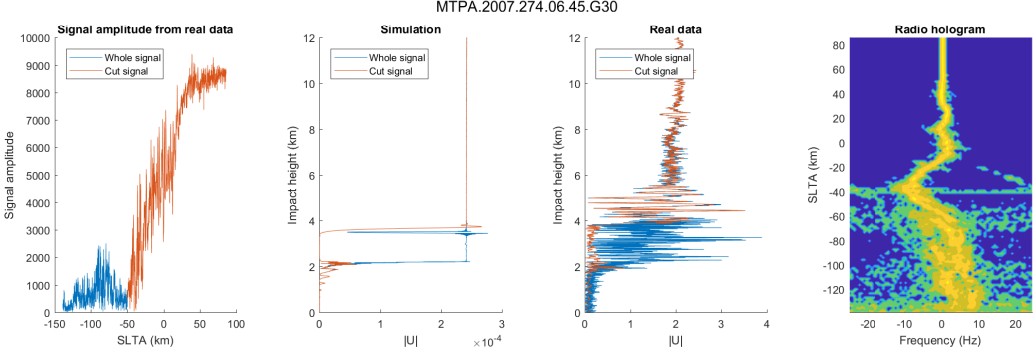

**Figure 10.** An event classified into category 3. $SLTA_{min}$ is at approximately $-85$ km, and the signal is truncated at $SLTA = -50$ km. The reflection spike is located at an impact height of 1928 m, and $a_{min}$ is 2200 m.

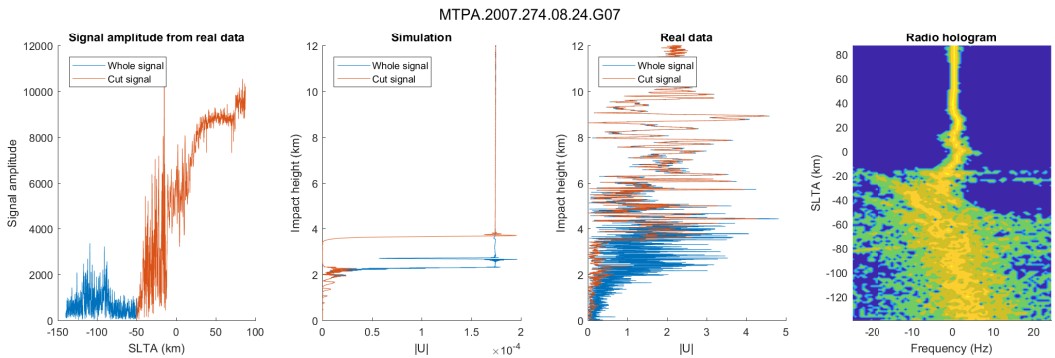

**Figure 11.** An event classified into category 3. $SLTA_{min}$ is at approximately $-99$ km, and the signal is truncated at $SLTA = -50$ km. The reflection spike is located at an impact height of 2363 m, and $a_{min}$ is 2301 m.

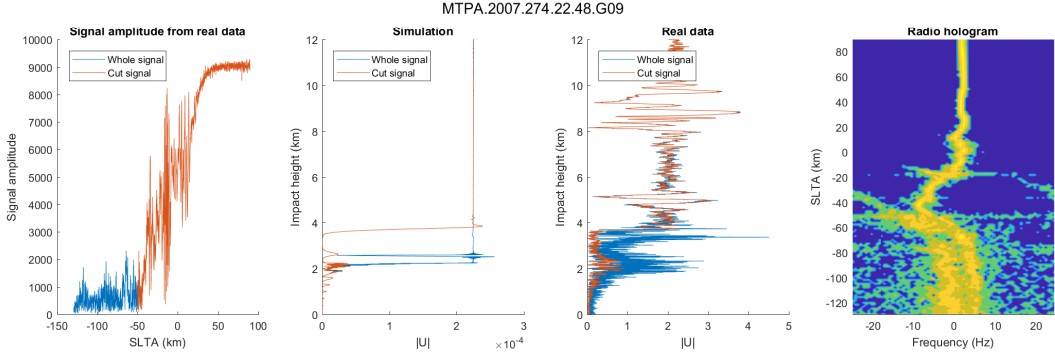

**Figure 12.** An event classified into category 4. $SLTA_{min}$ is at approximately $-103$ km, and the signal is truncated at $SLTA = -50$ km. The reflection spike is located at an impact height of 1994 m, and $a_{min}$ is 2232 m.

# 7 Conclusions

The results in this paper show that the signatures of surface reflections in GNSS RO measurements can be significantly enhanced when using the PM method by processing only an appropriately selected segment of the received signal. We can then identify reflection signatures even in cases where they are normally obscured by the direct signal's influence on the PM amplitude. This signature enhancement is demonstrated by simulations and confirmed with MetOp-A measurements. For further validation of the reflections, radio holographic images are provided for comparison. For the cases presented, clear and sharp reflection spikes in the PM amplitude are corroborated with clear reflection patterns in their corresponding radio holograms. Weaker or less distinct spikes have more noise in their corresponding radio holograms.

The events containing reflected signals presented here are hand-picked to illustrate that the approach can be useful for identifying reflections in real measurements. There are still questions that can only be answered by larger volumes of data, e.g. if the PM amplitude can possibly be used to retrieve information related to the reflected signal. Furthermore we observe that the reflection spikes vary greatly in shape. Additionally, simulated reflection spikes always occur very close to the surface impact parameter, however in real measurements this is not always the case. The reason for these variabilities needs to be investigated. At this point we have only analyzed events over water, since we expect those reflections to be much clearer and numerous compared to events over land.

When comparing the PM amplitudes of real and simulated signals, some interesting discrepancies were found. First, the characteristic dip in $|U|$ associated with a region of sharp gradient in the refractivity could not be identified in the real data, being completely drowned in noise, or simply not present. Second, the levels of noise in the $|U|$ function tends to increase close to the Earth surface. As no noise was added to the simulations, it is not clear if this is due to instrument noise, or atmospheric disturbances. Third, peculiar oscillatory features at quite distinct heights were seen in the real data but not at all in the simulated. These features are not understood at present, but it is likely that they have to do with horizontal gradients in the refractive index, and with aliasing of the reflected signal onto the direct signal due to low sampling rate. What horizontal gradients do to the PM amplitude has not been investigated, but it is likely that it will decrease the amplitude, and cause additional noise, perhaps resulting in these peculiar oscillations. Regarding the aliasing of reflected signals it is quite clearly seen in some of the radio holograms that the reflected signal is still strong when it has been aliased and again approaches the direct signal in frequency. This may cause a degradation of the PM method (the integral over the stationary phase point) at the specific impact height that belongs to the direct signal at the point where the direct and reflected aliased signal match in frequency.

# 8 Data availability

The data from occultation events is collected from the COSMIC Data Analysis and Archive Center (CDAAC) web interface, found at the URL http://cdaac-www.cosmic.ucar.edu/cdaac/index.html.

## Appendix A: Phase matching for reflected rays

It is not obvious that the phase matching method should work for reflected rays without modifications, but we will show that it does, under the assumption of reflections taking place on a perfectly smooth surface. First we will review the method used for phase matching of direct (non-reflected) rays, and then we will show that the result is the same for reflected rays. For the full details of the phase matching method the reader is referred to (Jensen et al., 2004).

### A1  Direct rays

Under the assumption of a spherically symmetric atmosphere we can use Bouger's rule

$$rn(r)\sin\phi = a, \tag{A1}$$

where $r$ is the distance from the Earth centre of curvature, $n$ the refractive index, $\phi$ the angle the ray makes with the radial vector, and $a$ the impact parameter. A ray is emitted from the GNSS satellite (at $r_G$) with angle $\phi_G$, being smaller than $\pi$. The ray makes its closest approach to the Earth when $\phi = \pi/2$. The ray then exits the atmosphere and is received at the LEO satellite (at $r_L$) with the angle $\phi_L$, being larger than $\pi$. The total bending of the ray (measured positive towards the Earth) is given by the bending angle

$$\alpha(a) = -2a \int_a^\infty \frac{1}{\sqrt{r^2 n(r)^2 - a^2}} \frac{d\ln n}{dr} dr. \tag{A2}$$

The optical path length for the ray is given by the integral over the refractive index along the path of the ray

$$S = \int n(r) ds, \tag{A3}$$

where the term $ds$ is an infinitesimal length along the ray. Under the spherical symmetry assumption the integral can be recast in a very attractive form, viz

$$S(t,a) = \sqrt{r_L(t)^2 - a^2} + \sqrt{r_G(t)^2 - a^2} - 2a^2 \int_a^\infty \frac{1}{\sqrt{r^2 n(r)^2 - a^2}} \frac{d\ln n}{dr} dr - 2 \int_a^\infty \sqrt{r^2 n(r)^2 - a^2} \frac{d\ln n}{dr} dr. \tag{A4}$$

The last term is connected to the bending angle in the following way

$$\int_a^\infty \alpha(a') da' = -2 \int_a^\infty \sqrt{r^2 n(r)^2 - a^2} \frac{d\ln n}{dr} dr, \tag{A5}$$

which can be verified by taking the derivative wrt $a$ on both sides. Using also the definition for the bending angle (Eq. (A2)) we can write

$$S(t,a) = \sqrt{r_L(t)^2 - a^2} + \sqrt{r_G(t)^2 - a^2} + \alpha(a)a + \int_a^\infty \alpha(a')da'. \tag{A6}$$

The impact parameter is generally connected to a certain point in time, and certain values for $r_L$ and $r_G$, in a complicated way.

5    Whatever this connection may be the angles in the system must fulfill

$$\theta(t) + \phi_G + \phi_L - \pi = \alpha, \tag{A7}$$

where $\theta$ is the separation angle between the satellites. We can rewrite this using Bouger's rule

$$\theta(t) + \arcsin(a/r_L(t)) + \arcsin(a/r_G(t)) - \pi = \alpha. \tag{A8}$$

For every value of $a$ there will be a corresponding value for $t$. In that sense one could write the optical path length as a function

10    of $t$ only, viz

$$S(t) = \sqrt{r_L(t)^2 - a(t)^2} + \sqrt{r_G(t)^2 - a(t)^2} + \alpha(a(t))a(t) + \int_{a(t)}^\infty \alpha(a')da'. \tag{A9}$$

In the phase matching method we perform an integral for each value of a given impact parameter $a_g$ where we wish to find the bending angle. The signal is given by

$$u(t) = |u(t)| \exp(ikS(t)), \tag{A10}$$

15    where $k_0$ is the wavenumber, and $i = \sqrt{-1}$. We subtract a geometrical model for the ray and form an integral as

$$U(a_g) = \int_{t_{min}}^{t_{max}} |u(t)| \exp(ik(S(t) - S_g(t, a_g))), \tag{A11}$$

where $t_{min}$ and $t_{max}$ are the start and stop times of the signal, and the geometrical ray is given by

$$S_g(t, a_g) = \sqrt{r_L(t)^2 - a_g^2} + \sqrt{r_G(t)^2 - a_g^2} + a_g\alpha(a_g) =$$
$$= \sqrt{r_L(t)^2 - a_g^2} + \sqrt{r_G(t)^2 - a_g^2} + a_g\left(\theta(t) + \arcsin(a_g/r_L(t)) + \arcsin(a_g/r_G(t)) - \pi\right), \tag{A12}$$

and the terms in the brackets are the bending angle from Eq. (A8). The integral will get its main contribution from the point where there is a stationary phase point, characterized by

$$\frac{d}{dt}\left(S(t) - S_g(t, a_g)\right) = 0. \tag{A13}$$

The time derivative of $S$ is given by

$$\frac{dS}{dt} = \frac{1}{r_L(t)}\frac{dr_L}{dt}\sqrt{r_L(t)^2 - a(t)^2} + \frac{1}{r_G(t)}\frac{dr_G}{dt}\sqrt{r_G(t)^2 - a(t)^2} + a(t)\frac{d\theta}{dt}. \tag{A14}$$

Likewise, the time derivative of $S_g$ is

$$\frac{dS_g}{dt} = \frac{1}{r_L(t)}\frac{dr_L}{dt}\sqrt{r_L(t)^2 - a_g^2} + \frac{1}{r_G(t)}\frac{dr_G}{dt}\sqrt{r_G(t)^2 - a_g^2} + a_g\frac{d\theta}{dt}. \tag{A15}$$

Hence, the stationary phase point occurs where $a(t_g) = a_g$. At that point the difference in optical path length becomes

$$S(t_g) - S_g(t_g, a_g) = \int\limits_{a_g}^{\infty} \alpha(a')da', \tag{A16}$$

10   and the phase matching integral is given by

$$U(a_g) = C(t_g)\exp\left(ik\int\limits_{a_g}^{\infty}\alpha(a')da'\right), \tag{A17}$$

where $C(t_g)$ is an amplitude factor depending on the signal amplitude and phase in the region around the stationary phase point. The bending angle as a function of impact parameter is thus found by taking the derivative of the phase of the function $U$ with respect to $a_g$, i.e.

$$\alpha(a_g) = -\frac{1}{k}\frac{d\angle U(a_g)}{da_g} \tag{A18}$$

## A2   Reflected rays

For rays suffering reflection the Bouger's rule still applies, but the ray never reaches the point where $\phi = \pi/2$. Instead the ray is reflected at the point where $r = R_E$, where $R_E$ is the Earth radius of curvature. Using the definition $R_E n(r_E) = a_E$ we find the angle the ray makes with the radial vector at reflection to be

20
$$\phi_E = \arcsin(\frac{a}{a_E}). \tag{A19}$$

Here we naturally assume that $a < a_E$, otherwise the ray would never reach the surface and be reflected. Since we assume the surface to be completely smooth, the radial vector is parallel to the surface normal, and since the incidence angle with respect to the surface normal is equal to the reflected ray angle with respect to the surface normal, we find that the ray angle after reflection is

$\quad \phi'_E = \pi - \phi_E.$ (A20)

We conclude that the ray suffers a negative bending of $\pi - 2\phi_E$ radians due to the reflection. The total bending angle for a reflected ray is therefore given by

$$\alpha(a) = -2a \int_{a_E}^{\infty} \frac{1}{\sqrt{r^2 n(r)^2 - a^2}} \frac{d\ln n}{dr} dr - \pi + 2\phi_E.$$ (A21)

The integral for the optical path length becomes more complicated (although the derivation is straightforward)

$\quad S(t) = \sqrt{r_L(t)^2 - a(t)^2} - \sqrt{a_E^2 - a(t)^2} + \sqrt{r_G(t)^2 - a(t)^2} - \sqrt{a_E^2 - a(t)^2}$

$$- 2a(t)^2 \int_{a_E}^{\infty} \frac{1}{\sqrt{r^2 n(r)^2 - a(t)^2}} \frac{d\ln n}{dr} dr - 2 \int_{a_E}^{\infty} \sqrt{r^2 n(r)^2 - a(t)^2} \frac{d\ln n}{dr} dr.$$ (A22)

We can rewrite slightly using eqs. A21 and A19

$$S(t) = \sqrt{r_L(t)^2 - a(t)^2} - \sqrt{a_E^2 - a(t)^2} + \sqrt{r_G(t)^2 - a(t)^2} - \sqrt{a_E^2 - a(t)^2}$$

$$a(t) \left[ \alpha(t) + \pi - 2\arcsin\left(\frac{a(t)}{a_E}\right) \right] - 2 \int_{a_E}^{\infty} \sqrt{r^2 n(r)^2 - a(t)^2} \frac{d\ln n}{dr} dr,$$ (A23)

the derivative wrt time yields

$$\frac{dS}{dt} = \frac{r_L(t)\dot{r}_L(t) - a(t)\dot{a}(t)}{\sqrt{r_L(t)^2 - a(t)^2}} + \frac{r_G(t)\dot{r}_G(t) - a(t)\dot{a}(t)}{\sqrt{r_G(t)^2 - a(t)^2}} + \dot{a}(t)\left[ \alpha(t) + \pi - 2\arcsin\left(\frac{a(t)}{a_E}\right) \right] +$$

$$+ a(t)\dot{\alpha}(t) + 2a(t)\dot{a}(t) \int_{a_E}^{\infty} \frac{1}{\sqrt{r^2 n(r)^2 - a(t)^2}} \frac{d\ln n}{dr} dr,$$ (A24)

where dot signifies derivative wrt time. The last term can be rewritten using A21

$$\frac{dS}{dt} = \frac{r_L(t)\dot{r}_L(t) - a(t)\dot{a}(t)}{\sqrt{r_L(t)^2 - a(t)^2}} + \frac{r_G(t)\dot{r}_G(t) - a(t)\dot{a}(t)}{\sqrt{r_G(t)^2 - a(t)^2}} + \dot{a}(t)\left[ \alpha(t) + \pi - 2\arcsin\left(\frac{a(t)}{a_E}\right) \right] +$$

$$+ a(t)\dot{\alpha}(t) + \dot{a}(t)\left[ -\alpha(t) - \pi + 2\arcsin\left(\frac{a(t)}{a_E}\right) \right].$$ (A25)

The expression simplifies to

$$\frac{dS}{dt} = \frac{r_L(t)\dot{r}_L(t) - a(t)\dot{a}(t)}{\sqrt{r_L(t)^2 - a(t)^2}} + \frac{r_G(t)\dot{r}_G(t) - a(t)\dot{a}(t)}{\sqrt{r_G(t)^2 - a(t)^2}} + a(t)\dot{\alpha}(t). \tag{A26}$$

The geometrical conditions in Eq. (A8) are still valid, viz

$$\alpha(t) = \theta(t) + \arcsin(a(t)/r_L(t)) + \arcsin(a(t)/r_G(t)) - \pi. \tag{A27}$$

Taking the derivative wrt time yields

$$\dot{\alpha}(t) = \dot{\theta}(t) + \frac{\dot{a}(t) - a(t)\dot{r}_L(t)/r_L(t)}{\sqrt{r_L^2(t) - a^2(t)}} + \frac{\dot{a}(t) - a(t)\dot{r}_G(t)/r_G(t)}{\sqrt{r_G^2(t) - a^2(t)}}. \tag{A28}$$

Inserting this expression into Eq. (A26) we get

$$\frac{dS}{dt} = \frac{\dot{r}_L(t)}{r_L(t)}\sqrt{r_L(t)^2 - a(t)^2} + \frac{\dot{r}_G(t)}{r_G(t)}\sqrt{r_G(t)^2 - a(t)^2} + a(t)\dot{\theta}(t), \tag{A29}$$

which is the very same expression as Eq. (A14). Hence, the stationary phase point again occurs where $a(t_g) = a_g$. At this point
we have

$$S(t_g) - S_g(t_g, a_g) = -2\sqrt{a_E^2 - a_g^2} + a_g\pi - 2a_g\arcsin\left(\frac{a_g}{a_E}\right) - 2\int_{a_E}^{\infty}\sqrt{r^2n(r)^2 - a_g^2}\frac{d\ln n}{dr}dr. \tag{A30}$$

This is the term that appears in the phase of the phase matching function $U(a_g)$. Taking the derivative with respect to $a_g$ leads
to

$$\frac{d}{da_g}\left(S(t_g) - S_g(t_g, a_g)\right) = 2a_g\int_{a_E}^{\infty}\frac{1}{\sqrt{r^2n^2(r) - a_g^2}}\frac{d\ln n}{dr}dr + \pi - 2\arcsin\left(\frac{a_g}{a_E}\right), \tag{A31}$$

which is the negative of the bending angle for a reflected ray as given in Eq. (A21) Consequently the phase matching method
works in the exact same way for direct and reflected rays. It should be stressed that these derivations are only valid when the
Earth surface can be considered smooth. When the surface is not smooth the incoming ray will change impact parameter upon
reflection. Due to this the expression for the optical path length becomes a function of the old and new impact parameter,
and the simple geometrical ray model used in the phase matching method cannot lead to a stationary phase point. This is
basically a case of multipath in the impact parameter domain. It may be argued though that this is of little consequence for real
measurements since the occultation measuring instrument will not record signals that deviate too strongly from direct rays, as

they quite rapidly become heavily Doppler shifted with increasing reflection angle. For this reason reflected signals will only be seen at impact parameters that are very close to the value at the Earth surface. These rays are of grazing incidence, and under such circumstances the surface may always be considered as flat (Beekmann and A, 1963).

*Author contributions.* J. Rasch, T. Sievert, A. Carlström and M. I. Pettersson designed the study and T. Sievert performed the simulations and processing. T. Sievert prepared the manuscript with contributions from all co-authors. J. Rasch performed the calculations for the reflection model and wrote the appendix.

*Acknowledgements.* This research was supported by The Swedish National Space Board and The Knowledge Foundation (KKS).

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
