# Peer review of "Analysis of reflections in GNSS radio occultation measurements using the phase matching amplitude"

_Atmospheric Measurement Techniques, 2017_

## Referee Comment (RC1) · Anonymous Referee #1 · 28 Jul 2017

Review amt-2017-216
Sievert et al.

The manuscript seeks to prove that the phase matching amplitude technique applied to GNSS radio-occultation (GNSS RO) signals can be used for detection of GNSS signals reflected off the Earth surfce and interfere with the direct radio-link.

The topic is very interesting and this new technique to detect presence of reflected signals in RO events has potential to be easily implemented. Nevertheless, the manuscript is rather poor, written in a style and degree of detail and rigourosity which is closer to an internal report than a peer-review article. The authors claim the technique is a good detector, but this cannot be claimed when tested in only 10 cases, and especially if no assessment is done about the false positives. Moreover, they do not provide any way for the reader to cross-check whether these 10 cases do present indeed reflected signals, or not.

The authors present a forward model for the relationship between reflected bending and impact parameter as original, but this operator is already given in Gorbunov et al. 2016 (which points to the original source, a ROM SAF internal report). Authors must be aware of this, as in another location of the manuscript they cite Gorbunov et al. 2016.

The manuscript only mentions the canonical transform as an alternative way of detecting reflected signals in RO, while the ROM SAF is providing a list with of the order of millions of COSMIC RO events flagged with presence or not of reflected signals detected with a different technique (support vector machine). The authors are fully aware of this technique, as they do cite (for another reason) the ROM SAF report where the technique is described. By the way, the technique has been validated with several thousands of RO events, and clear assessments have been made not only of the positive detections but on the false positives, too.

---

## Referee Comment (RC2) · Anonymous Referee #2 · 9 Aug 2017

The paper presents a technique to identify surface reflection signatures inside Radio Occultation simulated/real signals. Even if the subject is interesting, it is already investigated (as pointed out also in the Introduction) and several methods to flag occultations when a reflection signature is observable, are already available and are already implemented. The shown technique is based on applying the Phase Matching technique to a somehow truncated signal which removes the noisy contribution coming from atmospheric multipath in the lower and deeper part of the occultation.

The paper does not present all the details, thus it is difficult to understand whether the strategy is effective or not. There is a kind of evidence with the results presented through the 10 examples but, in any case, it would be nice to have an independent validation considering other well consolidated techniques which are quite easy and

straightforward to be implemented (visual inspection of the radio holographic / frequency spectra).

Also because I'm rather convinced that what is shown by the authors (maybe this is what they really wanted to show) is not really the entire signature of reflection on the RO signal (to be used for some geophysical analysis), but the diffraction effect of the Earth's surface on the signal. The reflection signature should appear for grazing incidence in the real signal (up to when the doppler departure from the atmospheric doppler is not too big) for longer time (i.e. wider impact height) intervals, not only as a spike. In all the examples shown by the author there is always a spike at around the lowest impact height (which corresponds to the ellipsoidal surface modelling the Earth's surface). I suspect that the spike is always around at a distance of one Fresnel zone from the surface (where the diffraction effect on the signal amplitude is maximum). It is clear that the shown spike is related in some way to the interaction with the Earth's surface and it will allow to detect the presence of the Earth's surface in the occultation event (which might be also an useful information), but it does not reveal the presence of a reflection signature useful for geophysical analysis. By properly cutting the signal, such spike will probably always be found. Instead, reflection signatures will really depend on the surface scattering properties, and might appear or might remain hidden in the atmospheric signal signature.

This should be clearly addressed in the paper (in the abstract, discussion, conclusion), which should be entitled like "Detection of Earth's surface in GNSS RO measurements using phase matching".

Some other specific and technical comments are provided here below.

"Specific Comments"

**17: "In many RO... signal". Not sure what does it mean. Of course the atmosphere is**

always sounded above the place where the signal is reflected. Clarify

**24 and below: Data used to obtain Fig 1 are not described properly. Is the U function based on the simulated u(t) or on the measured one on METOP-A? Are the plots the U(a) amplitudes (Eq 1) evaluated when the matching signal is the one for reflected rays described in Annex A2? At #27 you are referring to a certain model. A model of what? The model of the geometrical optical path or the models allowing you to draw the SLTA(ha) for the direct (blue plot) or reflected (red plot) rays? Unclear**

**4: where n(RE) is taken? Is the colocated ECMWF refraction index profile? An exponential profile as addressed in #9, page 4?**

**15: Using the information of the Centre of Curvature is not necessary because you need to exploit "more accurate values of impact height". You need this reference system translation to fulfil the local spherical symmetry assumption.**

**21: "simpler geometry of the WOP". Please elaborate more this sentence.**

The entire Sect 5 should be more clearly addressed. It is quite confusing

**6: If the signal tracking is "lost", you cannot have any u(t), thus you cannot have any amplitude(U). Why are you referring to realistic instrument behaviour here (unless you can provide evidence that you can have an amplitude(U) spike in a realistic occultation signal when the tracking is lost).**

**8: Why are you saying "if tracking of the signal..." if Fig 2 is based on simulated occultation? Are you using also an instrument simulator (which provides an idea of the instrument tracking behaviour) together with the WOP?**

**9: Again on Fig1. Here you are referring to an exponential profile. Of what? Refraction index? What are you showing in Fig 1? The effects of SLTA(ha) for n(R) based on an exponential profile? See also the comment provided for #4 at Page 3.**

**9-10: "This shows... direct rays". The sentence is not clear. Do you want to mean that you can have reflection signatures also at higher SLTAs? This is of course true, up to when the associated Doppler will be within the receiver bandwidth. And thanks to this you can maybe do some geophysical analysis on the reflected signal. That's why I'm saying that what you called "reflection" spike (#13) is not the reflection signature you are dealing with here. In my view this reflection "spike" is the diffraction effect of the Earth's surface.**

**1: not clear at all which is the truncation strategy. Please add a sentence defining the strategy you used.**

**13: what does it mean that all occultation of that day have a reflection signature? Why this should depend on the "day"? Is the reflection signature the spike you was able to identify or is it because the reflection flag in the data was set to 1?**

Page 6/7/8: all the plots: Why the amplitude(U) dynamic is so different between simulated (from 0 to 10ˆ-4) and realistic (to 0 from 4) data? Measuring Unit are missing...

Page 11:

**8: this follows geometrical considerations. Also the Bouger's rule which is becomes an easy trigonometrical formulation for n = 1 (rsin(fi) = a).**

"Technical corrections"

Page2

**3: processing instead of receiving**
**10: address the Annexes content**

**14: k or k0? Is this the wave number in vacuum associated to the GNSS carrier frequency?**

**1: "orbital radii for the satellite are fixed" means "circular orbits"?**

---

## Author Comment (AC1) · 27 Sep 2017

On behalf of myself and my co-authors I want to thank the referees for their valuable comments. I attach to this post a .zip-file containing two PDF documents - one containing responses to every comment made by the referees, and one revised manuscript with marked-up changes in accordance to the comments we received.

Please also note the supplement to this comment: https://www.atmos-meas-tech-discuss.net/amt-2017-216/amt-2017-216-AC1-supplement.zip

---

## Author Response (AR1)

**Reviewer #1**

- Comment: The authors claim the technique is a good detector, but this cannot be claimed when tested in only 10 cases, and especially if no assessment is done about the false positives. Moreover, they do not provide any way for the reader to cross-check whether these 10 cases do present indeed reflected signals, or not.

- Response: The intention is not to present a fully developed detector, but rather to highlight that reflections appear in the PM amplitude. To make this clearer, we changed the title from "Detection" to "Analysis", and avoid referring to this as a "detection method". We added radio holographic images of the events we present as a means of validating the existence of reflection.

**Reviewer #1**

- Comment: The authors present a forward model for the relationship between reflected bending and impact parameter as original, but this operator is already given in Gorbunov et al. 2016

- Response: The relationship between reflected bending angle and impact parameter was never intended to be presented as our original contribution. We have clarified by adding appropriate references.

**Reviewer #1**

- Comment: The manuscript only mentions the canonical transform as an alternative way of detecting reflected signals in RO, while the ROM SAF is providing a list [...] (support vector machine) [...]

- Response: A more thorough description of the publications we refer to has been added to the introduction.

**Reviewer #2**

- Comment: […] in any case, it would be nice to have an independent validation considering other well consolidated techniques which are quite easy and straightforward to implement (visual inspection of the radio holographic / frequency spectra).

- Response: Radio holographic images have been added to the figures in the results section for validation. The trails going into negative frequencies indicate that the events contain reflections, and noisier radio holograms correspond to noisier phase matching amplitudes.

**Reviewer #2**

- Comment: […] but [the spike] does not reveal the presence of a reflection signature useful for geophysical analysis. By properly cutting the signal, such spike will probably always be found.

- Response: We maintain that the spike does indeed reveal the presence of a reflection signature – Fig. 1 in the paper motivates this. The reflected signal is present everywhere where there is a direct signal (except for the very deep direct signals), but only the lower parts of the reflected signal is picked up by the instrument. The direct and reflected rays merge at the impact parameter (and SLTA) corresponding to the Earth surface. At this point the direct signal is diffracted by the presence of the Earth. By truncating the signal at a much higher SLTA we remove the diffracted ray. The reason for the spike-like appearance of the reflection in the impact parameter domain is also explained by figure 1. Where it is seen that a large interval in SLTA for the reflected ray is compacted into a rather small interval in impact parameter. This has been elaborated in Sect. 5.

**Reviewer #2**

- Comment: Page 1#17: "In many RO… signal". Not sure what does it mean. Of course the atmosphere is always sounded above the place where the signal is reflected. Clarify

- Response: This was an unclear sentence, the point was to say that the signal is often reflected. It has been changed to be clearer.

**Reviewer #2**

- Comment: Page 2#24 and below: Data used to obtain Fig 1 are not described properly. Is the U function based on the simulated u(t) or on the measured one on METOP-A? Are the plots the U(a) amplitudes (Eq 1) evaluated when the matching signal is the one for reflected rays described in Annex A2? At #27 you are referring to a certain model. A model of what? The model of the geometrical optical path or the models allowing you to draw the SLTA(ha) for the direct (blue plot) or reflected (red plot) rays? Unclear

- Response: The model we refer to is the model of impact height vs SLTA, shown by the overlayed plots red and blue in Fig. 1. These plots are described by equations (2) and (3) and based on data from co-located ECMWF profiles. The black curves are |U| plots generated by measured MetOp-A data, which we have clarified. Appendix A2 serves to show that the PM operator admits reflected rays as well as direct ones, so the U functions shown in the figure are for different segments of the measured signal.

**Reviewer #2**

- Comment: Page 3#4: where n(RE) is taken? Is the colocated ECMWF refraction index profile? An exponential profile as addressed in #9, page 4?

- Response: n(RE) is taken from the colocated ECMWF profile. This information is added. In the last version of the manuscript we referred to an exponential profile. This is incorrect and has been removed.

**Reviewer #2**

- Comment: Page 3#15: Using the information of the Centre of Curvature is not necessary because you need to exploit "more accurate values of impact height". You need this reference system translation to fulfil the local spherical symmetry assumption.

- Response: This has been corrected.

**Reviewer #2**

- Comment: Page 3#21: "simpler geometry of the WOP". Please elaborate more this sentence.

- Response: We were referring to a two-dimensional geometry with a stationary GNSS transmitter. This has been elaborated in Sect. 4.

**Reviewer #2**

- Comment: Page 4: The entire Sect 5 should be more clearly addressed. It is quite confusing.

- Response: Sect 5 has been rewritten.

**Reviewer #2**

- Comment: Page 4#6: If the signal tracking is "lost", you cannot have any u(t), thus you cannot have any amplitude(U). Why are you referring to realistic instrument behaviour here (unless you can provide evidence that you can have an amplitude(U) spike in a realistic occultation signal when the tracking is lost).

- Response: Since the U function is an integral over the entire u(t), there cannot be "tracking" for any single point in U. Thus, U is defined for any impact parameter a, and the amplitude can tell us if a specific value of a corresponds to a ray or if it is just noise. The time span in which we have u(t) is considered the time when the instrument tracks the signal. What we mean when we say "tracking is lost" is that beyond that value of t we have no signal. We have added an explanation of how, in the WOP, we can mimic the measurement results of the instrument tracking down to a specific SLTA. This was added to Sect. 4.

**Reviewer #2**

- Comment: Page 4#8: Why are you saying "if tracking of the signal…" if Fig 2 is based on simulated occultation? Are you using also an instrument simulator (which provides an idea of the instrument tracking behaviour) together with the WOP?

- Response: What we mean to say is that the has been simulated down to a certain SLTA. By changing separation angle parameter in the WOP, we can mimic the measurement results that would be produced from an instrument tracking to lower or higher SLTA. A clarification of this was added to Sect 4.

**Reviewer #2**

- Comment: Page 4#9: Again on Fig1. Here you are referring to an exponential profile. Of what? Refraction index? What are you showing in Fig 1? The effects of SLTA(ha) for n(R) based on an exponential profile? See also the comment provided for #4 at Page 3.

- Response: Any reference to an exponential profile is a mistake and has been removed.

**Reviewer #2**

- Comment: Page 4#9-10: "This shows… direct rays". The sentence is not clear. Do you want to mean that you can have reflection signatures also at higher SLTAs? This is of course true, up to when the associated Doppler will be within the receiver bandwidth. And thanks to this you can maybe do some geophysical analysis on the reflected signal. That's why I'm saying that what you called "reflection" spike (#13) is not the reflection signature you are dealing with here. In my view this reflection "spike" is the diffraction effect of the Earth's surface.

- Response: Yes, what we mean is that reflected rays are received at higher SLTAs than the direct rays diffracted by the Earth's surface. If we process the whole signal however, the contribution of these rays overlap at the same impact height. We have added clarifying remarks in Sect 5 about this.

**Reviewer #2**

- Comment: Page 5#1: not clear at all which is the truncation strategy. Please add a sentence defining the strategy you used.

- Response: The truncation strategy is to cut the signal well above SLTAmin, and apply a tapering window to avoid artifacts. We do this to remove the direct rays corresponding to the bottom of the atmosphere. We elaborate on this in Sect. 5, and we explain for each of the ten measurements in their figure captions where we truncate and which SLTA we need to stay above.

**Reviewer #2**

- Comment: Page 5#13: what does it mean that all occultation of that day have a reflection signature? Why this should depend on the "day"? Is the reflection signature the spike you was able to identify or is it because the reflection flag in the data was set to 1?

- Response: In referring to "this day", we meant to refer to "the data we have looked at", since we looked at one day's worth of data. Our point was that for every $|U|$ we came across that looked a certain way, there was reflection. As this was not a clear wording, we changed it to make the point come across clearer: *"When the measurements are sufficiently deep, and the noise level of $|U|$ is low, there are very clear reflection spikes.".* This is what we seek to highlight with the category 1 cases we present.

**Reviewer #2**

- Comment: Page 11#8: this follows geometrical considerations. Also the Bouger's rule which is becomes an easy trigonometrical formulation for n = 1 (rsin(fi) = a).

- Response: We corrected this accordingly.

- The title and abstract were revised to clarify that we do not present a complete detection method, but rather an analysis of RO data.

- The introduction was revisedto more fully explain the work referenced, and minor changes were made to use clearer wordings.

- Section 2 was revised in a minor way to clarify some of the notation used in the paper.

- Section 3 was revised to more clearly explain figure 1, and to emphasize that we do not present the relationship between reflected bending angle and impact parameter as our own result.

- Section 4 was minorly revised to be clearer, and a paragraph describing the added radio holographic images was added.

- Section 5 was rewritten to explain more clearly why the reflection signatures differ from the signatures from the lowest, direct rays.

- Section 6 was revised to also explain the added radio holographic images.

- The conclusions were revised to reflect the title change and to take into account the added radio holographic images. Furthermore the conclusions were elaborated to address the large amount of noise we see in the analysis of MetOp-A data.

- Radio holographic images made from the corresponding measurements were added to figures 3-12, as well as more details about the truncation points.

[revised manuscript text omitted]

---

## Author Response (AR2)

- In accordance to issue A), information about the impact height of the spike and amin has been added to the figure captions of each presented case in the results.

- In accordance to issue B), more thorough mathematical evidence has been added, and the sign error has been addressed.

- All the editorial technical comments have been addressed as suggested, unless they are mentioned below.

- About Figure 2: The comment asks that it use the same simulations as Figure 1. Figure 1 in fact uses real data, whereas Figure 2 uses a simulation to show the general idea of truncation, and the need for a proper window. As such we do not deem it necessary to use the same data.

- About the negative peaks mentioned on page 6: The plots with simulated data (second from the left) all have dramatic negative spikes where a typical profile without sharp gradient would be completely smooth. These occur between 2-4 km depending on where there is a sharp gradient in the profile. We maintain that these spikes are clearly visible in the *simulated* data, however it can not be distinguished in the real measurements.

[revised manuscript text omitted]